

# Geology and vegetation control landsliding on forest-managed slopes in scarplands

Daniel Draebing[1,2], Tobias Gebhard[1], Miriam Pheiffer[1]

[1]Chair of Geomorphology, University of Bayreuth, Bayreuth, 95447, Germany
5  [2]Department of Physical Geography, Utrecht University, Utrecht, 3584 CB, Netherlands

*Correspondence to*: Daniel Draebing (d.draebing@uni-bayreuth.de)

**Abstract.** Landslides are important agents of sediment transport, cause hazards and are key agents for the evolution of scarplands. To analyse geologic and vegetation control on landsliding, we investigated three landslides in the Franconian scarplands. We used geomorphic mapping, soil analysis, electrical resistivity and a mechanical stability model to quantify the 10 stability state of the landslides. Furthermore, we mapped tree distribution, quantified rooted area and root tensile strength to assess the influence of vegetation on shallow landsliding. Our results show that landslides are deep-seated incorporating rotational and translational movement with sliding along a geologic boundary between permeable Rhätolias sandstone and impermeable Feuerletten clays. Despite low slope angles, landslides could be reactivated when high pore pressures could develop due to geologic conditions. In contrast, shallow landsliding is controlled by vegetation. Our results show that rooted 15 area is more important than species dependent root tensile strength and limited to the upper 0.5 m of the surface due to geologically controlled unfavourable soil conditions. Due to low slope inclination, root cohesion can stabilize landslide toes or slopes undercut by forest roads, independent of potential soil cohesion, when tree density is sufficient dense. Forest management currently adapts forests to climate change by diversifying tree species and introducing European beech, which would increase slope stability when sufficient rooted area is reached. Forestry activities should aim to keep a certain tree 20 density to enable sufficient root cohesion that prevent landslide activity between harvesting or adaption periods. In summary, geological conditions in scarplands favour landslide activity and influence vegetation control on landslide activity, which suggest a combined forest and hazard management should be applied to prevent future landsliding.

## 1 Introduction

Large parts of continental land surfaces are of sedimentary origin with sediments deposited in terrestrial or marine 25 environments (Duszyński et al., 2019). Depending on the tilting of sedimentary layers these scarplands are characterized by horizontal layers, forming plateaus delineated by escarpments or by tilting layers resulting in cuestas (Young et al., 2000; Duszyński et al., 2019). The sedimentary layers possess different rock strength and usually the top layers forming the escarpment or cuesta have higher strength than underlying layers (Duszyński et al., 2019). The scarplands are eroded by mass movements that can differ between frontscarps, where sediment layers dip into the slope, and backscarps characterized by



sedimentary layers dipping out of the slope (Schmidt and Beyer, 2003; Duszyński et al., 2019). At frontscarps, landsliding in form of rockfall (e.g. Glade et al., 2017) or large landslides (e.g. Jäger et al., 2013) are abundant. In contrast, landsliding processes are characterized by cambering (Hutchinson, 1991), block gliding (Young, 1983), lateral spreading (Spreafico et al., 2017) or sliding processes (Pain, 1986; Schmidt and Beyer, 2003) at backscarps. Sliding processes can occur even on low-inclined hillslopes if clay layers are forming the shear plane (Skempton, 1964; Chandler, 2000; Bromhead, 2013). These

landslides can be classified based on type of movement, shear surface, depth of failure plane, type of material and velocity (Varnes, 1978). Depending on depth of failure plane, landslides can be further divided into deep-seated or shallow landslides, where shallow landslides are characterized by material <2 m deep moving downslope in a flowing, sliding or complex type of movement (Sidle and Bogaard, 2016; Vergani et al., 2017). Shallow landslides are often nested on large landslides reworking landslide deposits (e.g. Pánek et al., 2013). Forests can affect shallow landsliding mechanically and hydrologically (Vergani

et al., 2017). They can reduce soil moisture by interception and evaporation, suction and transpiration as well as infiltration and subsurface flow (Sidle and Bogaard, 2016; Vergani et al., 2017). Mechanically, forests can reinforce soil by roots (Wu, 1984; Phillips et al., 2021), roots and stems can induce buttressing (Vergani et al., 2017) and anchoring and trees can increase normal force on slopes (Selby, 1993).

Forests cover 35 % of Europe's total land area (Ministerial Conference on the Protection of Forests in Europe, 2020) including

scarplands and are an important resource for construction material, non-wood forest products, pulp and paper and energy production (Mubareka et al., 2016). In Germany, 75 % of all trees are spruces (25.4 %), pines (22.3 %), beeches (15.4 %) or oaks (10.4 %, Thünen-Institut, 2013). Norway spruce (*Picea abies*) and Scots pine (*Pinus sylvestris*) are the most important economic coniferous species in Europe (Caudullo et al., 2016; Houston Durrant et al., 2016b) and European beech (*Fagus sylvatica*), pedunculated (*Quercus robur* R.) and sessile oak (*Quercus petraea* (Matt.) Liebl.) are important species for

silviculture (Eaton et al., 2016; Houston Durrant et al., 2016a), therefore, their distribution is affected by forest management activities. In forest management, the protective function of forests has been considered for a long time in high mountain regions (Dorren et al., 2005; Bischetti et al., 2009). However, forestry is not only affected by landslide activity, which causes damage to roads and loss of timber (Sidle and Ochiai, 2006), but also has a considerable impact on slope stability through changing the characteristics of forests in sliding-prone areas (Phillips et al., 2021). Root reinforcement of slope stability declines after

logging operations (Schmidt et al., 2001; Vergani et al., 2017) and forestry roads enhance landsliding through undercutting slopes (Borga et al., 2005; van Beek et al., 2008). Changes in tree species composition and tree density have also an impact on the root reinforcement in forests (Roering et al., 2003; Genet et al., 2008). Climate change affects both forests (e.g. Seidl et al., 2017) and landslide activity (e.g. Crozier, 2010). Therefore, forest management efforts are required to adapt forests to changing environmental conditions (Bartsch and Röhrig, 2016) that in turn can affect future landslide occurrence.

The influence of vegetation on landslides has been intensely studied on steep slopes in the European Alps (Bischetti et al., 2009; Vergani et al., 2014), the Oregon Coast Range (Schmidt et al., 2001; Roering et al., 2003), Northern Italy (Borga et al., 2005; Schwarz et al., 2010b), New Zealand (Giadrossich et al., 2020) or China (Genet et al., 2008). However, little effort was conducted to understand the influence of vegetation on landsliding on lower-inclined hillslopes such as scarplands in Southern



Germany (e.g. Thiebes et al., 2014) or in the Flemish Ardennes (e.g. Van Den Eeckhaut et al., 2009), where geologic conditions
enable landsliding on low-inclined slopes. In this study, we aim to (1) quantify the geologic control on landslides occurring in
the Franconian scarplands in Southern Germany, and (2) quantify the influence of trees on slope destabilisation in a forest-
managed environment.

## 2 Study area

The research area is located in Northern Bavaria, Germany (Fig. 1A). Geologically, it is situated at the north-eastern margin
of the Franconian Alb, which is the backscarp part of the scarplands in south Germany that consists of sand-, clay- and
limestone of mostly Mesozoic age dipping gently to the East, Southeast and South (Kany and Hammer, 1985; Peterek and
Schröder, 2010). Tributaries of the Red Main River eroded deep valleys into the hillslopes that consist of Middle Triassic to
Lower Jurassic clay- and sandstones. The lower part of the hillslopes are formed by claystones called Feuerletten that are part
of the Trossingen Formation and were deposited during prolonged flooding events in the Middle Triassic (Emmert, 1977).
They are characterized by red violet, fine sandy clay- and clay-marlstones, which are weathered near surface into clay. The
clay minerals consist of smectite, sudoite and illite (Emmert, 1977) with high swelling potential (Wilfing et al., 2018),
impermeable and serving as an aquiclude (Boley Geotechnik, 2018). Silty and sandy lenses lead to inhomogeneities and highly
varying mechanical parameters (Wilfing et al., 2018). The Feuerletten are overlain by a sequence of sand- and claystones,
which are part of the Exter-Formation (Upper Triassic) and Bayreuth-Formation (Lower Jurassic). The strata are embraced as
Rhätolias and the sandstones form the escarpment in the scarplands. The Exter-Formation consists of a pronounced spatial
heterogeneous sequence of sandy and clayey deposits, which vary greatly in their thickness. The predominant dark-coloured
clays are characterised by the occurrence of montmorillonite and kaolinite (Emmert, 1977) with high swelling capacity that
can promote the formation of sliding surfaces (Wilfing et al., 2018). Intercalated quarzitic sandstone layers are predominant
fine- to coarse-grained with fluviatile cross bedding (Meyer and Schmidt-Kaler, 1996). The Bayreuth-Formation is formed by
a mostly massy, coarse-grained and light-coloured sequence of sandstones with a cross-bedding structure and intercalated
subordinate clayey lenses (Emmert, 1977). The Rhätolias strata serves as aquifer over the Feuerletten clays that significantly
reduce the hydraulic permeability and are interpreted as sliding planes of abundant landslides (Kany and Hammer, 1985). The
intense fracturing of the sandstones, due to the tectonic strain near the Franconian line, allows water to penetrate into the soil
and leads to the formation of sliding surfaces along the clayey layers (Wilfing et al., 2018). The climate in the research area is
warm-moderate with an annual precipitation around 719 mm and an annual temperature about 8.9 °C for the period 1991 to
2020 (DWD Climate Data Center, 2022b, a).







**Figure 1: (a) Location of the research area at the Franconian Alb (source: Bayerisches Landesamt für Digitalisierung, Breitband und Vermessung). (b) mapped landslides based on Bayerisches Landesamt für Umwelt (2015) and own mapping including investigated landslides (source: Bayerisches Landesamt für Digitalisierung, Breitband und Vermessung). (c) Frequency-magnitude relationship of landslides.**

## 3 Methods

### 3.1 Geomorphic and geologic characterisation

On regional scale (Fig. 1B), we revised the existing landslide inventory by Bayerisches Landesamt für Umwelt (2015) for our research area and mapped additional landslides based on a Digital Elevation Model (DEM) with a resolution of 1 m. To analyse



the role of geology, we derived the boundary between Rhätolias and Feuerletten from existing geological maps (Bayerisches Geologisches Landesamt, 1955, 1977, not yet published). We created a frequency-magnitude relationship based on our landslide inventory (Fig. 1C).

On local scale, geomorphic mapping was conducted in the field on three landslides with focus on landslide-induced landforms and geomorphic maps were created in ArcGIS 10.7.1 (Fig. 2). We used a longitudinal transect that started in un-affected terrain above the headscarp, went across the landslide down to or across the stream. Along this transect, we conducted 1 m long Pürckhauer soil coring with 25 to 35 m spacing to analyse the soil texture according to Ad-hoc-AG Boden (2005).

## 3.2 Electrical Resistivity Tomography

Electrical resistivity tomography (ERT) is a standard technique to investigate landslides (Perrone et al., 2014). The technique is well suited to differentiate landslide thickness in lithologies producing contrasting resistivities such as mudstone and sandstone (Chambers et al., 2011; Uhlemann et al., 2017) or loess and tertiary sands (Van Den Eeckhaut et al., 2007). To investigate three landslides, we applied ERT along the 360 to 400 m longitudinal transects using an ABEM Terrameter LS2. We measured a Wenner array with 5 m spacing of electrodes, which enabled a penetration depth of 60 to 70 m. For data

processing, we used a robust least-squared inversion in Geotomo Res2DInv (Loke and Barker, 1995). Model results showed a low root mean square (RMS) error between 5.3 and 5.4% for Putzenstein and Weinreichsgrab and an increased RMS error of 12.1% at Fürstenanger, which are comparable to RMS values of previous investigations (e.g. Van Den Eeckhaut et al., 2007; Perrone et al., 2014). The uncertainty analysis revealed highest uncertainties near to the surface (Figure S1) and uncertainties between 1 and 5 % at our area of interest, the potential shear plane. A minimum and maximum analysis showed that ERT

results are consistent (Figure S2) and data processing was not affecting the results. We used virtual 1D-ERT boreholes to identify the shear plane depth (Siewert et al., 2012) and applied a minimum, mean and maximum shear plane depth scenarios (Figure S3) for our landslide stability model.

## 3.3 Tree mapping and influence on stability

We mapped trees up to a lateral distance of 5 m along our approximately 400 m long ERT transects. Tree mapping included

location and tree species of trees larger than 4 m. Dead and cut trees were excluded as the influence of roots on cohesion decays with ongoing decomposition (Vergani et al., 2014; Zhu et al., 2020). At different positions along our transects, we selected individual free-standing trees with a diameter at breast height (DBH) between 30 and 45 cm to minimize variations in root stability due to different growth and age (Deljouei et al., 2020). At 15 trees, we dug 0.5 m wide and 0.5 m deep soil pits in 0.8 m distance downslope of the stem (Ji et al., 2012) of trees reflecting the three main tree species of the area: Norway

spruce, Scots pine, and European beech. To determine the root area ratio (RAR), we took photos of each soil pit, georeferenced the photos in ArcGIS 10.7.1 using tie points designated by measurement tapes in both vertical and horizontal directions,



mapped every visible root and determined location and diameter (Vergani et al., 2014; Hales and Miniat, 2017). Roots with a diameter <1 mm were excluded to avoid uncertainties and roots with a diameter >10 mm were neglected as they do not contribute to the tensile strength of roots due to their stiffness (Bischetti et al., 2009; Vergani and Graf, 2016). To analyse RAR

depending on depth, the profile wall was divided into 10 cm depth intervals and RAR was calculated:

$$RAR = a_r = \frac{\sum_{i=1}^{i} A_{ri}}{A} \tag{1}$$

with the root cross-sectional area $A_r$ is

$$A_r = \frac{\pi}{4} d^2, \tag{2}$$

root diameter $d$ and $A$ the area of each 0.1 m segment of the soil pit. To measure root tensile strength, root samples with

different diameters and a minimum length of 10 cm were extracted for Scots Pine, while tensile strength power laws are available for Norway spruce and European Beech (Genet et al., 2005; Bischetti et al., 2009). Sampled roots were watered for one hour to compensate for different moisture content and to ensure tensile strength measurement under wet conditions (Hales et al., 2013). Tensile strength measurements were performed applying the set-up by Hales et al. (2013) using a spring scale (G&G OCS-XY) with 0.01 kg resolution suspended with a rope on a horizontal branch. The roots were clamped using a grip

tong and vertically pulled downwards until breakage using a pincer. The weight at breakage was recorded and the root diameter at breakage measured using a digital calliper (Preciva) with a resolution of 0.01 mm. Only tests with a root breakage near the middle were used for the statistical analysis (Genet et al., 2005; Bischetti et al., 2009). The force at failure $FF_r$ was calculated from the recorded weight $w$:

$$FF_r = w\, g \tag{3}$$

with $g$ is the gravitational acceleration. The root tensile strength $T_r$ was calculated following previous studies (Schmidt et al., 2001; Genet et al., 2005):

$$T_r = \frac{FF_r}{A_r}. \tag{4}$$

A power-law between root tensile strength and root diameter $d$ can be established:

$$T_r(d) = \alpha d^{-\beta} \tag{5}$$

with α and β are empirical constants depending on species. Power law parameters used in the analysis ranged from 18.10 $d^{-0.72}$ ($r^2 = 0.52$) for Norway spruce to 41.57 $d^{-0.98}$ ($r^2 = 0.65$) for European beech (Bischetti et al., 2009).

The total root tensile strength $t_r$ across the profile wall can be calculated incorporating the RAR for ith root diameters ranging from 1 to 10 mm (Bischetti et al., 2009):

$$t_r = \sum_{i=1}^{n} (T_r a_r)_i \tag{6}$$

with $n$ is the number of roots per diameter class $i$. The root cohesion $c_r$ was calculated following previous studies (Waldron, 1977; Wu, 1984; Schmidt et al., 2001):

$$c_r = (cos\alpha\, tan\emptyset + sin\alpha)\, t_r = k' t_r \tag{7}$$





with $\alpha$ is the angle of root deformation from the vertical angle by shearing (see Fig. 2 in Schmidt et al., 2001) and $\varphi$ as the angle of internal friction of the soil. For roots with $40° < \alpha < 70°$ and $25° < \phi < 40$, k' is around 1.2 (Wu et al., 1979). Applying Eq. 7 with α similar to Wu et al. (1979) to the angle of internal friction for Feuerletten (Table 1) k' is around 1, which is in accordance to Bischetti et al. (2009). To consider for non-simultaneous root breakage, the correction factor k'' in the order of 0.5 was applied following Bischetti et al. (2009):

$$c_r = k'k''t_r. \tag{8}$$

Root cohesion can be differentiated into basal and lateral root cohesion (Schwarz et al., 2010a). The basal root cohesion is characterized by roots crossing the shear plane of landslide at a depth $z$. Following Bischetti et al. (2009), Eq. 6 can be adapted to

$$c_{bas}^z = \left(k'k'' \sum_{i=1}^{N}(T_r a_r)_i\right)_z \tag{9}$$

with $N$ is the number of roots at a given depth.

The lateral root cohesion results from roots intersecting the vertical plane of a detachment scarp:

$$c_{lat}^z = \sum_{j=1}^{M}\left[k'k''\left(\sum_{i=1}^{N}(T_r a_r)_i\right)_j \frac{\Delta z_j}{z}\right] \tag{10}$$

with $M$ is the number of depth classes of thickness $\Delta z_j$.

The total root cohesion is the sum of basal and lateral root cohesion

$$c_r^z = c_{bas}^z + c_{lat}^z. \tag{11}$$

**3.4 Landslide stability model**

Mechanical strength parameters of Feuerletten and Rhätolias were quantified using approximately 90 circular, direct and triaxial tests on materials from the Thurnau landslide (Fig. 1B) affecting the highway (Boley Geotechnik, 2018; Wilfing et al., 2018).





**Table 1.** Strength parameters according to Boley Geotechnik (2018) measured at A70 (Fig. 1b) with cohesion $c'_s$, angle of friction $\phi$ and residual angle of friction $\phi'_R$.

| Geology | Soil | $c'_s$ (kPa) | $\phi$ (°) | $\Phi'_R$ (°) |
|---|---|---|---|---|
| *Rhätolias* | Clay/silt layers | 24.4 – 99.4 | 15.8 – 30.7 | 10.0 – 27.1 |
| | Sand-gravel | 0.1 – 6.6 | 23.1 – 38.1 | |
| | Sand (baked) | 82.9 – 102.1 | 24.0 – 28.6 | 20.5 |
| | *Median* | 48 | 23.1 | 13.8 |
| | *25% quantile* | 24.4 | 18.9 | 10.4 |
| *Upper Feuerletten* | Silty clay (stiff) | 49.0 – 126.0 | 13.4 – 24.1 | |
| | Silty clay (soft) | 11.3 – 45.9 | 13.4 – 26.4 | 8.4 |
| | Silty clay (baked) | 17.5 – 28.9 | 18.8 – 25.7 | |
| | Claystone | 94.9 | 20.1 | |
| | *Median* | 47.5 | 19.0 | |
| | *25% quantile* | 27.9 | 16.0 | |

To assess the reactivation of identified landslides, we used the method of slices following Fellenius (1936) and calculated the
factor of safety $F$:

$$F = \frac{\sum_{i=1}^{n}[c'_{si}\, l_i + (W_i\, cos\beta_i - \mu_i\, l_i)\, tan\emptyset_i]}{\sum_{i=1}^{n}[W_i\, sin\beta_i]} \tag{12}$$

with $i$ is the number of slices, $c'_s$ is the soil cohesion, $l$ is the base length of each slice and $\beta$ is the angle of failure plane (Selby, 1993). For each slice, $W_i$ needs to be calculated:

$$W_i = \gamma_s z_i B_i \tag{13}$$

with $\gamma_s$ is the specific weight of soil assuming a soil density 1800 kg m$^{-3}$ and $B_i$ is the width of each slice. The depth $z_i$ and the
angle of the failure plane $\beta_i$ of each slice, we derived from the ERT and applied an upper, mean and lower depth to incorporate
uncertainties associated with the applied geophysical technique.

The pore water pressure $\mu_i$ is calculated for each slice:

$$\mu_i = \gamma_w\, m\, z_i\, cos^2\theta \tag{14}$$

with $\gamma_w$ is the specific weight of water assuming a water density of 997 kg m$^{-3}$ and $\theta$ is the slope angle. The slope angle $\theta$ was
derived from the DEM. As the saturation is unknown, we scaled saturation using

$$m = \frac{H}{z} \tag{15}$$

with $H$ is the height of the water table. We calculated stability scenarios from no saturation (m = 0) to full saturation (m = 1;
Table 2). Where F<1, the slope is in condition of failure, while slopes with F>1 are considered as stable (Selby, 1993).






**Table 2.** Factor of safety scenarios for the reactivation of the entire landslide. L, m, u refer to lower, mean and upper shear plane depth scenario.

| Scenario | $z$ (m) | $c'_s$ (kPa) | $\Phi$ (°) | $m$ (m/m) |
|---|---|---|---|---|
| 1 (blue) | l, m, u | 28.6 | 8.4 | 0 - 1 |
| 2 (yellow) | l, m, u | 8.5 | 8.4 | 0 - 1 |
| 3 (green) | l, m, u | 0 | 8.4 | 0 - 1 |

To assess the susceptibility of shallow landsliding at undercut areas or at landslide toes, we used the infinite slope model by Skempton and De Lory (1957):

$$F = \frac{c'_s + c_r^Z + (\gamma_s - m\gamma_w)z\cos^2\theta\tan\emptyset}{\gamma_s z\sin\theta\cos\theta}. \tag{16}$$

We calculated the factor of safety for cohesion scenarios ranging from no cohesion to 10 kPa assuming full saturated conditions (m=1) and a residual angle of friction of 8.4° to test if root cohesion would be sufficient to stabilize the soil.

## 4 Results

### 4.1 Geomorphology of the landslides

In our research area, 125 landslides were identified (Fig. 1b) with an area ranging from 745 m² to 320,220 m² (Fig. 1c). Around 95 % of the landslides are crossed by the Rhätolias-Feuerletten boundary and all landslides follow no expositional pattern (Fig. 1b). The cumulative number of landslides plotted against area showed a typical distribution with a decreasing number of landslides with increasing landslide size (Fig. 1c). We mapped three of the largest ten landslides in detail. All three landslides are characterized by the location of the headscarp within the Rhätolias formation. The Putzenstein landslide has a 710 m long headscarp and a length of up to 310 m resulting in an area of approximately 150,000 m² (Fig. 2a). Several secondary scarps and depressions are located above the headscarp without indicators of recent movement (e.g. bent trees). Within the northern most headscarp part, we observed roots that were under tension (Fig. 3a-c). The landslide area is very hummocky and the landslide front has a height between 1 and 2 m. The Weinreichsgrab landslide is characterized by a 490 m long headscarp, a length between 110 m and 330 m and an area of 120,000 m² (Fig. 2b). The headscarp exposed partially 10 - 15 m vertical sandstones (Fig. 3d) in contrast to 45° inclined slopes (Fig. 3e). The landslide area is hummocky and the front characterized by tilted and bent trees (Fig. 3f). The Fürstenanger landslide has a 490 m long headscarp (Fig. 2c) with a 10-15 m high 45° inclined slope (Fig. 3g-h). The landslide is up to 290 m long and comprises an area of 150,000 m². The upper third of the landslide area is hummocky and the landslides developed into a straight slope ending at the river (Fig. 2c). At Putzenstein



landslide, fine and silty sand was abundant in the first 300 m transect length. At 325 m transect length, reddish clays underlay a 0.2 m thick sand layer. A similar pattern was visible at Weinreichsgrab with silty and fine sand were abundant until 320 m transect length, where clays underlain a 30 to 50 cm thick sand layer. At Fürstenanger, silty and fine sand occurred until 180

m transect length. First layers of clay overlain by silty sand were observed at 24 to 40 cm depth between 181 and 206 m transect length. Clays were abundant at the surface from 236 m on but usually overlain by a 20-40 cm thick organic and silt layer.





**Figure 2: Geomorphic maps of the landslides at (a) Putzenstein, (b) Weinreichsgrab and (c) Fürstenanger (DEM source: Bayerische Vermessungsverwaltung).**



Figure 3: Photos of (a-b) lateral roots under tension located at the headscarp and (c) at a secondary scarp of the Putzenstein landslide. (d) Rhätolias boulders at the headscarp, (e) overview of the headscarp and (f) tilted trees at the toe of the Weinreichsgrab landslide. (g-h) Overview about the headscarp and (i) the toe of the Fürstenanger landslide.






## 4.2 Landslide thickness

The Putzenstein landslide was characterized by three high resistant cells located at transect lengths between 5 and 70 m, 70 to 195 m and 232 to 315 m with resistivities up to 4,000 Ohm m (Fig. 4a). The cells' thickness ranged from 18 m at the beginning

of the transect to 7 m at the lower part of the transect. The underlying areas below these cells had low and contrasting resistivities that enabled a clear differentiation from the near surface areas (Fig. 4a and S3a-c). Between the high-resistant cells, the ERT revealed two low-resistant bodies between 195 and 220 m as well as between 315 and 330 m transect length with low contrast to underlying areas (Fig. S3d). The Weinreichsgrab landslide was characterized by high-resistant areas near the surface until transect length 320 m (Fig. 4b) with underlying low-resistant areas from 9 m depth on (Fig. S3e-g). The high-

resistant areas were differentiated into three cells in clear contrast to underlying low-resistant areas, while the lower part of the transect from 320 m on showed low resistivities between 20 and 70 Ohm m with low contrast to underlying areas (Fig. S3h). The Fürstenanger landslide revealed heterogeneous near surface conditions with alternating high-resistant and low-resistant areas (Fig. 4c). At transect length 50 to 110 m, an up to 15 m thick high resistant body was located. From 110 m on, the transect was characterized by alternating low- and high resisting cells and a more or less consistent zone of contrasting resistivities at

4 m depth (Fig. S3j, l). This pattern was disturbed at 180 m transect length, where areas of higher resistivities dipped 45° into the slope resulting in a 10-12 m thick zone of contrasting low and high resistivities (Fig. S3k).

Earth **Surface**
**Dynamics**
Discussions

Figure 4: Geoelectric models and landslide forms at (a) Putzenstein, (b) Weinreichsgrab and (c) Fürstenanger. F highlights location of forest roads.



### 4.3 Trees and roots

The tree mapping results on all transects showed high spatial variability of tree species composition. The Putzenstein landslide showed a clearing with seedlings of different species including Scots pine, European silver fir and European larch above the headscarp and until transect length 75 m (Fig. 5a). From 75 m on, the tree cover got denser with young Norway spruces and European beeches. Between 180 m and 235 m transect length, the trees were characterized by young Norway spruces and European beeches of mixed ages. Norway spruces became dominant from 235 m on and grew in form of a dense thicket

between the first forest road at 260 m and the second forest road at 325 m. Below the second forest road, an abrupt change occurred and trees were characterized by Norway spruces, European beeches and Scots pines of different ages. Above the headscarp of the Weinreichsgrab landslide (Fig. 5b), old trees stood in an open high forest, mixed with the grouped regeneration of Norway spruce. Between 40 and 150 m transect length, young European beeches and European silver firs grew with Scots pines that added to the regeneration. From 150 m on, the species mixed with older Norway spruce trees until the forest road at

225 m. Below the forest road, Norway spruces were dominant and young Norway spruces grew in thickets. From 320 m on, many Norway spruces and few European beeches occurred, but were misaligned or dead. The area above the headscarp of the Fürstenanger landslide and from transect length 50 to 120 m was characterized by Scot pines (Fig. 5c). From transect length 120 m on, Norway spruce became the dominant species and grew in form of a thicket from 150 m until the forest road at 170 m. Below the forest road, Norway spruces with a few European beeches and Scots Pines occurred until 280 m transect length,

while Norway spruces were dominant at the landslide toe.



Figure 5: **Mapped trees with height above 4 m in up to 5 m distance to the ERT transect.**

From 160 root tensile strength tests, 27 tests showed a root breakage in the middle and were used to develop a tensile strength
root diameter relationship. This relationship is characterized by an exponential decrease of tensile strength with increasing root
diameter (14.22 d$^{-1.13}$, r²=0.55; Fig. 6). Roots were restricted to the upper 0.5 m for Scots pines and Norway spruces and to 0.4
m for European beeches. The RAR showed no differences between Rhätolias or Feuerletten. For Norway spruce, mean root





area ratio decreased from the surface to 0.5 m with 0.19 and 0.2 % at 0 to 0.2 m depth, 0.04 % at 0.2 to 0.4 m depth and 0.005 % between 0.4 and 0.5 m depth (Fig. 7a). Only the depth between 0.1 and 0.2 m showed a variation between the sites.

Scots pines showed a similar RAR trend with RAR values between 0.16 and 0.19 % between 0 and 0.2 m, 0.01% between 0.2 and 0.3 m, 0.04 % and 0.11% between 0.3 and 0.5 m depth (Fig. 7b). The variability was highest from 0 to 0.3 m depth between measurement locations. European beeches revealed a similar RAR depth pattern, however, with increased magnitudes and variability. Mean RAR decreased from 0.42 % between 0 and 0.1 m depth to 0.15 and 0.12 % between 0.1 and 0.4 m depth (Fig. 7c). Root cohesion revealed similar depth patterns as RAR. For Norway spruce, mean root cohesion showed 12 to

12.4 kPa for 0 to 0.2 m depth and decreased to 3.6 kPa between 0.2 and 0.3 m, 2.6 kPa between 0.3 and 0.4 m and 0.7 kPa between 0.4 and 0.5 m depth (Fig. 7d). Scots pine revealed lower root cohesion. Mean root cohesion increased from 4.3 kPa between 0 and 0.1 m to 6 kPa between 0.1 and 0.2 m depth (Fig. 7e). With increasing depth, mean root cohesion fluctuated between 1.6 and 2.3 kPa. European beeches showed the highest root cohesion magnitude and variability. Mean root cohesion decreased from 23.6 kPa for the upper 0.1 m to a range between 8.4 and 12.5 kPa for depths between 0.1 and 0.4 m (Fig. 7f).


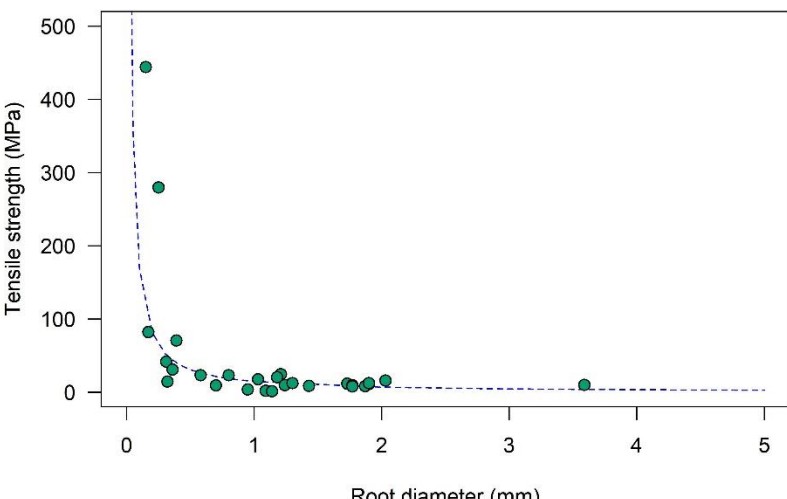

**Figure 6: Tensile strength plotted versus root diameter for Scots pine.**





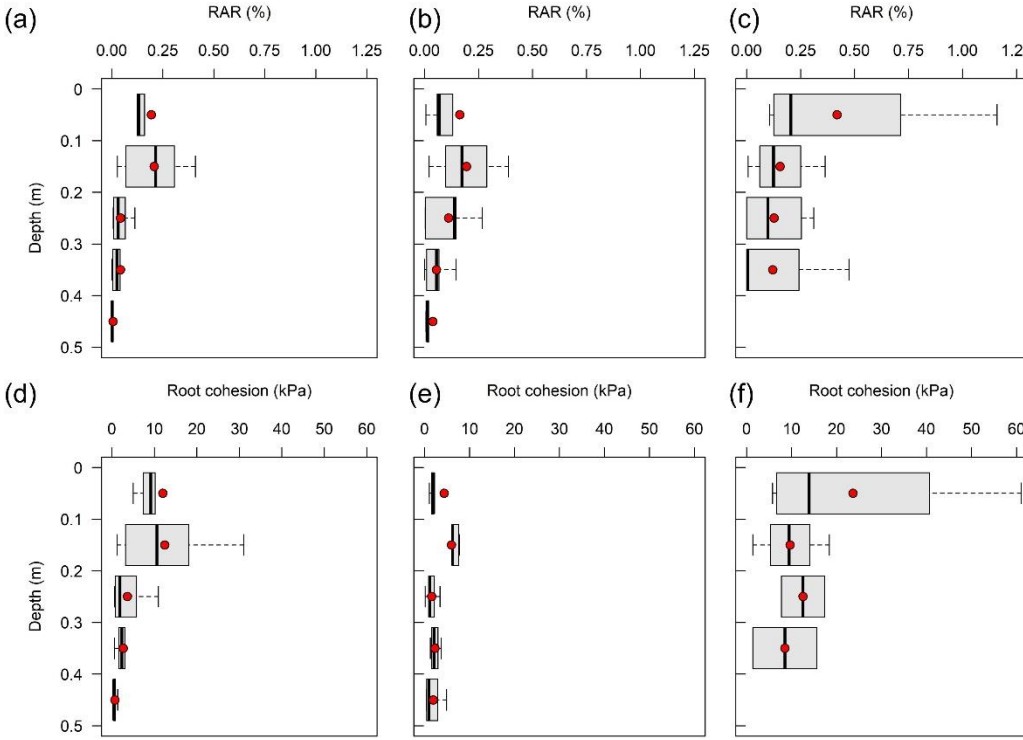

**Figure 7: Root area ratio plotted against depth for (a) Norway spruce, (b) Scots pine and (c) European beech. Root cohesion plotted against depth for (d) Norway spruce, (e) Scots pine and (f) European beech. Red dots highlight mean RAR or root cohesion.**

## 4.4 Landslide stability analysis

### 4.4.1 Landslide stability scenarios of the entire slope

All three landslides revealed shear planes far below rooting depth and showed stable conditions with factor of safety (FoS) values above 1.7 when assuming a soil cohesion of 28.6 kPa (Fig. 8). Assuming a residual cohesion of 8.5 kPa resulted in FoS values all over 1 at all water levels at Putzenstein landslide (Fig. 8a). For Weinreichsgrab and Fürstenanger landslides, stability depended on the slice height scenario. The Weinreichsgrab landslide got instable between a saturation of 0.8 and 1.0 only in the upper slice height scenario (Fig. 8b). The Fürstenanger landslide undercut a factor of safety of 1 at full saturation for the mean and at 0.8 for the upper slice height scenario (Fig. 8c). Assuming no residual soil cohesion, the Fürstenanger landslide would be instable independent of saturation levels (Fig. 8c). The Putzenstein landslide would get instable between a saturation level of 0.55 and 0.85 depending on shear plane scenario (Fig. 8a). The Weinreichsgrab landslide would be instable for maximum shear plane scenario independent of saturation and for mean and minimum shear plane scenario above 0.45 (Fig. 8b). Assuming, full saturation would reduce soil cohesion to zero, all scenarios for all landslides would show a FoS below 1.





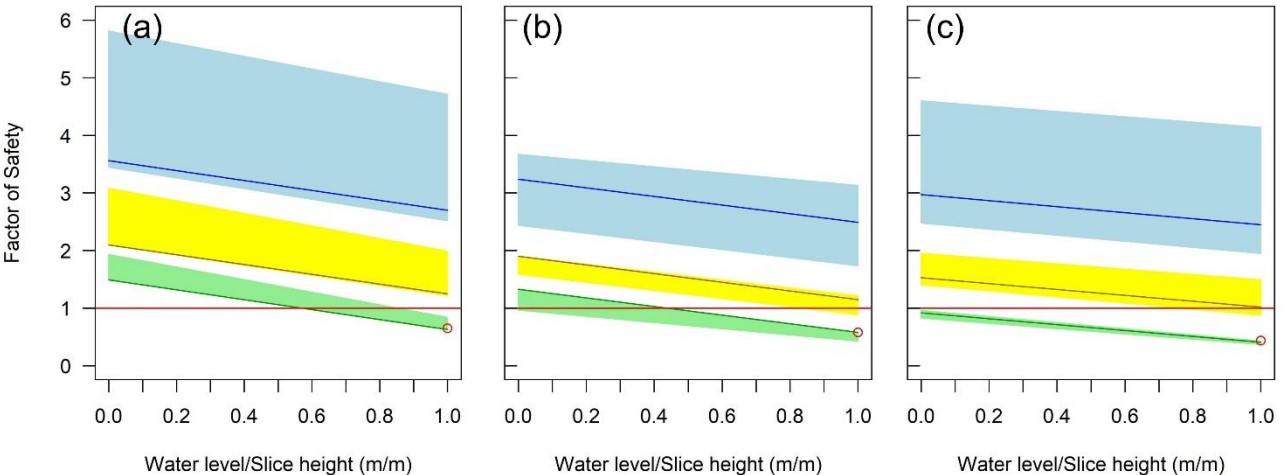

**Figure 8: Factor of safety models for the reactivation of the landslides at (a) Putzenstein, (b) Weinreichsgrab, and (c) Fürstenanger. We assume an angle of internal friction of 8.4° and a cohesion between 28.6 kPa (blue scenario), 8.5 kPa (yellow) and 0 kPa (green). Calculations are based on a mean shear plane depth (line) and minimum and maximum shear plane depth (rectangle).**

330

### 4.4.2 Landslide stability scenarios of slopes above road cuts and landslide toes

Root cohesion can act as basal root cohesion when penetrating the shear plane and as lateral root cohesion when anchoring the soil during scarp development. We calculated which minimum combined soil and root cohesion is necessary to prevent the occurrence of shallow translational landslides above road cuts and at landslide toes. All these locations are characterized by

335   Feuerletten and assuming a soil cohesion of 28.6 kPa or a residual soil cohesion of 8.5 kPa would result in stable conditions (Fig. 9). When the soil is oversaturated, soil cohesion can be zero. In this case, a minimum root cohesion between 0.8 kPa for shear planes at 0.3 m and 4.2 kPa for shear planes at 1.5 m would be sufficient to stabilize the soil above road cuts (Fig. 9a). At landside toes, root cohesion between 0.25 and 0.8 kPa are required to stabilize a potential landslide with a shear plane at 0.3 m depth and root cohesion between 1 and 3.4 kPa to stabilize landslides with shear planes of 1.5 m depth (Fig. 9b).

340



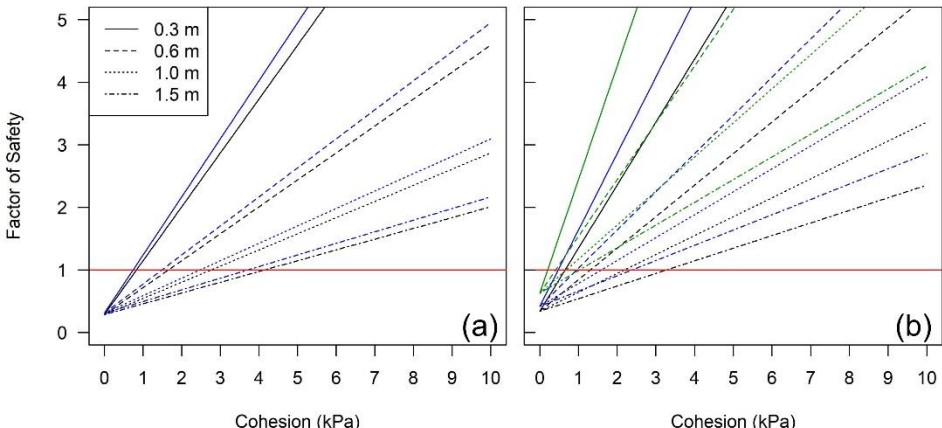

**Figure 9: Factor of safety for full-saturated conditions with a residual angle of friction of 8.4° plotted against cohesion scenarios ranging from no cohesion to 10 kPa for (a) translational landslides at road cuts and (b) landslide toes. Line style highlight the depth of shear plane ranging between 0.3 m and 1.5 m. Line colour in (a) refer to Putzenstein (black) with a slope angle of 13°, Weinreichsgrab and Fürstenanger (both blue) with slope angles of 12°. Line colour in (b) refer to Putzenstein (black) with a slope angle of 11°, Weinreichsgrab (blue) with a slope angle of 9° and Fürstenanger (green) with a slope angle of 6°.**

## 5 Discussion

### 5.1 Geologic control on landsliding

We observed 125 landslides in our research area with 95 % of the landslides are crossed by the Rhätolias-Feuerletten boundary, which suggests that Feuerletten play a key role in landsliding. Previous landslide inventories of Franconian Alb support this role of Feuerletten in the North-Bavarian scarplands, where Feuerletten were responsible for an inappropriate high part of landslides (Kany and Hammer, 1985). Kany and Hammer (1985) suggested that the shear plane developed at the border between permeable Rhätolias and impermeable Feuerletten resulting in translational and rotational landsliding or mostly in a combination of both. Furthermore, the authors assumed that most landslides were fossil but could be reactivated due to anthropogenic impacts as road cutting and forestry (Kany and Hammer, 1985).

The Putzenstein landslide revealed a hummocky topography (Fig. 2a) and the ERT showed three high-resistant cells with resistivities up to 4,000 Ohm m and a thickness between 7 and 18 m located above low-resistant bodies at transect length between 5 and 70 m, 70 to 195 m and 232 to 315 m (Fig. 4a). Pürckhauer drillings revealed fine and silty sand in the upper 1 m. We interpret these cells as dry Rhätolias above wet Feuerletten. The form of these cells and the hummocky topography indicate three rotational slabs. In between the lower high-resistant cells, low resistivities indicate a water-saturated rotational slab (Fig. 4a). The lower part of the landslide was characterized by a flat topography, low-resistant areas, and near-surface clay material. Therefore, we interpret this landslide part as a translational slide within the Feuerletten. The Weinreichsgrab landslide revealed a similar pattern of three high-resistant cells within hummocky terrain with near-surface silty sand followed by flat terrain with low resistivities and near-surface clay (Fig. 4b). These results indicate three rotational slabs and one





translational slab at the toe of the landslide. In contrast, the Fürstenanger landslide showed one high-resistant area in the upper part indicating a rotational failure (Fig. 4c). However, the major part of the landslide showed heterogeneous near-surface resistivities underlain by low resistivities in form of a straight slope indicating a translational landslide. The observed resistivity pattern was disturbed at 180 m transect length, where areas of higher resistivities dipped 45° into the slope resulting in a 10-12 m thick zone of contrasting low and high resistivities (Fig. S3 k). However, the topography showed no evidence of a

rotational slide, therefore, we interpret the resistivity pattern as an artefact of the measurement rather than an indicator for rotational movement.

Landslide prediction is complicated by uncertainties regarding material properties, such as soil cohesion and shear plane depth (Almeida et al., 2017). Electrical resistivity tomography enabled in most conditions the identification between of the shear plane due to high resistivity contrasts between Rhätolias and Feuerletten with an uncertainty depending on the resolution of

the tomography in the range of 2.5 m. Therefore, we established minimum, mean and maximum shear plane depth scenarios to propagate the uncertainty into our stability analysis. All scenarios were below the rooting depth of trees observed on the landslides, therefore, we used only soil cohesion in the landslide stability analysis. Further uncertainties rise from the wide range of material strength properties. Material properties were derived from the landslide affecting the highway (Thurnau in Fig. 1b). They showed a large variation between individual layers and within each layer of the Feuerletten (Table 1; Boley

Geotechnik, 2018; Wilfing et al., 2018), but were in the range of previous investigations (Kany and Hammer, 1985; Wiedenmann, 2019). We tested the reactivation of the entire landslide and used the residual internal angle of friction of 8.4° measured by Boley Geotechnik (2018). For soil cohesion, we used different scenarios. According to Skempton (1964), clay develops a residual soil cohesion of zero after shearing, however, laboratory tests by Ikari and Kopf (2011) indicate that a residual soil cohesion can re-develop in clays. Therefore, we used a mean cohesion of 28.6 kPa as upper scenario, a reduced

soil cohesion of 8.6 kPa (1/3 of the original value) as mean scenario and no soil cohesion as lower scenario (Table 2).

Our landslide stability analysis showed that all three landslides revealed stable conditions independent of saturation with FoS values above 1.7 when assuming a soil cohesion of 28.6 kPa (Fig. 8). When assuming a residual cohesion of 8.5 kPa, an FoS below 1 is not reached at Putzenstein landslide independent of water level (Fig. 8a), at Weinreichsgrab below saturation of 0.8 in the upper slice height scenario (Fig. 8b) and at Fürstenanger below 0.8 for the maximum and 1.0 for the mean shear plane

scenario (Fig. 8c). The development of high saturation in the sand layers of Rhätolias is unlikely as sand is very permeable. However, Rhätolias has impermeable clay layers (Boley Geotechnik, 2018) and tectonic-induced fractures can increase water infiltration through these clay layers (Wilfing et al., 2018). Water can move laterally at depth below the impermeable Feuerletten and Rhätolias clay layers and can cause hydrostatic pressures equal to high saturation levels (Rogers and Selby, 1980; Selby, 1993). Therefore, a reactivation of the entire landslide could be possible due to the geologic conditions of

alternating clay layers within the Rhätolias underlain by impermeable Feuerletten.

Assuming no residual soil cohesion as suggested by Skempton (1985), the Fürstenanger landslide would be instable independent of saturation level and shear plane scenario (Fig. 8b-c), while the Putzenstein and Weinreichsgrab landslide would get instable between a saturation level of 0.55 to 0.8 and 0 to 0.45 depending on shear plane scenario (Fig. 8a-b). However,





there are no indicators for instability on the landslides except tensed roots at Putzenstein (Fig. 3a-c) and bent or tilted trees at
Weinreichsgrab (Fig. 3f) that indicate soil creep or shallow landsliding but no reactivation of the entire landslides. Therefore,
we assume that the soil cohesion re-established and currently prevents slope failure under low-saturated conditions. Climate
change can increase the probability of extreme wet conditions in winter (Estrella and Menzel, 2013) that can cause in
combination with snow melt high pore pressures capable to trigger landsliding. To identify rainfall thresholds that enable
landslide triggering, detailed monitoring of hydrostatic pressures would be necessary (e.g. Wilfing et al., 2018).


## 5.2 Vegetation control on landsliding

Trees influence the landslide and in turn, the landslide influences the trees. Roots provide stability to landslide-prone slopes
and the influence of roots in stability depends on tensile strength and rooted area (e.g. Wu, 1984; Phillips et al., 2021). Based
on 27 tests we developed a tensile strength root diameter relationship for Scots pines, which is characterized by an exponential
decrease of tensile strength with increasing root diameter ($r^2$=0.55; Fig. 6). Therefore, relative tensile strength increases with
decreasing root diameters (Stokes et al., 2009) as thinner roots possess a higher cellulose content that provides additional
strength (Genet et al., 2005). The power law and the statistical degree is in the range of previous measurements (Genet et al.,
2005; Bischetti et al., 2009) and show only little difference between species (Genet et al., 2005; Hales, 2018). Our findings
suggest that rooted area has a higher influence on landslide stability than tree species-specific tensile strength.

Our RAR measurements showed that roots were restricted to the upper 0.5 m for Scot pines and Norway spruces and to 0.4 m
for European beeches (Fig. 7a-c). Within a species, RAR revealed no differences between topographic locations at the slope
or between Rhätolias or Feuerletten. The rooting depth was very low compared to pines and beeches occurring in the near-by
Frankenwald that showed rooting depth up to 1.2 m (Nordmann et al., 2009), however, lithology and soil conditions are
different, which seem to influence root properties more than species identity (Lwila et al., 2021). At upper slope location,
Rhätolias is abundant and characterized by high permeable sandy soil. In dry soils, trees usually develop deeper roots to reach
groundwater (Hoffmann and Usoltsev, 2001), however, the hard sandstone layers within the Rhätolias prevent deeper rooting.
In addition, sandy soils are less deeply warmed than fine-grained soils which results in shallower root growth (Kutschera and
Lichtenegger, 2002). At lower slope locations, clayey Feuerletten are abundant which resulted in combination with slope-
induced water flow in moist conditions. Moist aerated soils are characterized by extreme flat rooting (Stone and Kalisz, 1991;
Kutschera and Lichtenegger, 2002). Therefore, lithology and associated soil conditions in combination with topography-
controlled water flow resulted in low rooting depth. Consequently, basal root cohesion can only effect shallow landslides with
a shear plane below 0.4 or 0.5 m depth, respectively.

Our RAR measurements showed that root density decreases with depth and revealed twice-higher RAR values for European
beeches than Scots pines or Norway spruces (Fig. 7a-c). This pattern was observed at alpine field sites (Bischetti et al., 2009),
however, the authors observed much higher RAR values for all tree species, a further indication that local conditions at our
landslide slopes limited root growth. Root cohesion revealed similar depth patterns as RAR (Fig. 7d-e) with decreasing root
cohesion with depth as RAR has the strongest influence on the root cohesion magnitude (Mao et al., 2012).





We tested shallow landsliding with shear planes up to 1.5 m depth for slopes affected by forest road cuts and at landslide toes with clay material near the surface enabling high saturation (m=1). Slopes above forest road cuts were characterized by low

inclination between 11 and 12°, while landslide toes revealed even lower slope angles in the range of 6 to 9°. Assuming a shear plane depth of 0.3 m, slopes above road cuts and landslide toes would require a cohesion between 0.2 and 0.8 kPa (Fig. 9) to stabilize the slope. As root cohesion of Norway spruce, Scots pine and European beech between 0.3 and 0.4 m depth is above 1 kPa (Fig. 7d-f), root cohesion would be sufficient to stabilize the slope. However, species distribution, number and position have an influence on the occurrence of landslides (Roering et al., 2003), as the vegetation patterns always leave gaps with

lower root cohesion. Our investigated slopes above road cuts were characterized by a combination of European beech and Norway spruce at Putzenstein and Weinreichsgrab landslides (Fig. 5a-b), which grew dense enough to provide sufficient root cohesion to stabilize the slopes. Dense thickets of Norway spruce occurred on Fürstenanger slopes above road cuts and on all landslide toes (Fig. 5c) and provide high root density that would enable sufficient stabilization.

When shear planes exceed rooting depth, lateral root cohesion can have a stabilizing effect (Schwarz et al., 2010b) as already

indicated by tensed roots observed at Putzenstein (Fig. 3b). To stabilize shallow landslides with shear planes up to 1.5 m, our calculations showed that a cohesion between 1 and 4 kPa would be required (Fig. 9). As lateral root cohesion is the sum of root cohesion of rooted depth, all three investigated species would provide sufficient lateral root cohesion to stabilize the slope (Fig. 7d-f) independent of potential soil cohesion, when spacing of trees enable an entire cover of the slope. Sufficient tree cover is provided at landslide toes and at the slope above the road cut at Fürstenanger (Fig. 5c), where thickets of Scots pine

are abundant. Above road cuts at Putzenstein and Weinreichsgrab, European beeches occur that provide the highest calculated root cohesion (Fig. 7f). Despite the calculations suggest that lateral root cohesion should prevent shallow landsliding, tilted and bent trees especially at Weinreichsgrab (Fig. 3f) indicate the occurrence of soil creep and potential slow shallow landslide movement (Van Den Eeckhaut et al., 2009; Pawlik and Šamonil, 2018).

Forestry activities influence slope stability. Roots decay after forest cutting results in decreasing strength and RAR decreases

(Vergani et al., 2014; Vergani et al., 2016) already relevant one year after tree cutting (Sidle and Bogaard, 2016; Zhu et al., 2020). In addition to reduction of root cohesion by timber harvesting (Vergani et al., 2016) or small-scale logging (Bischetti et al., 2016), the harvesting process can result in soil erosion (Haas et al., 2020) and the construction of new forest access roads increases instability through slope fragmentation and altered drainage (Borga et al., 2005; van Beek et al., 2008). Forestry activity can induce gaps in the forest cover that would decrease the effect of lateral root cohesion (Cohen and Schwarz, 2017).

Therefore, forestry activity at slopes or above road cuts could decrease root cohesion sufficiently to trigger landslides in case of non-existing soil cohesion due to high saturation levels. Regeneration of young trees may already provide a considerable amount of root reinforcement but takes years to restore the original root cohesion (Sidle and Ochiai, 2006; Giadrossich et al., 2020).

Climate change will result in higher probability of dry summers and wet winters (Estrella and Menzel, 2013). On dry locations

as permeable Rhätolias sandstone, droughts can affect the growth of Norway spruce, Scots Pine and with less effect of European beech (Debel et al., 2021). In the research area, forest management aims to adapt tree composition to climate change





(Keenan, 2015). In detail, the forest service aims to reduce the number of Norway spruce and increase the percentage of European beech (personal communication by F. Maier). Our RAR and root cohesion data (Fig. 7) suggests that a species change towards European beech would increase root reinforcement on the slopes when a sufficient rooted area has been
developed. Furthermore, the forestry service will diversify the tree composition by planting European sliver fir (*Abies alba*), Norway maple (*Acer platanoides*), European alder (*Alnus glutinosa*), sessile oak (*Quercus petraea* (Matt.) Liebl.), pendunculate oak (*Quercus robur* R.), Silver birch (*Betula pendula* Roth) and downy birch (*Betula pubescens* Ehrh.) and diversify tree age (personal communication by F. Maier). Previous investigations on plant diversity showed that tree mixture had no influence on FoS as root tensile strength plays a minor role in stability (Genet et al., 2010). However, root strength
decreases with age (Sidle and Bogaard, 2016), therefore, a mixed age forest can prevent root strength decay as young trees can compensate the reduction of root strength of old trees. Our root cohesion data showed (Fig. 7) that lateral root cohesion is sufficient to stabilise slopes (Fig. 9), when tree distribution is dense enough to avoid gaps. Therefore, stability is more a factor of tree size and density (Genet et al., 2010), and forest management should aim to achieve a dense enough forest that provides sufficient lateral and basal root cohesion (Fig. 9) to avoid future shallow landsliding.

## 6 Conclusion

Scarplands are characterized by alternating sedimentary layers with different strength properties. In our study, we observed 125 deep-seated landslides that indicate a geologic control on landsliding by impermeable Feuerletten underlying more permeable Rhätolias. Detailed investigations on three landslides showed that shear planes occurred at depth to deep for tree roots, therefore, roots play no role for slope stabilization. The wide range of potential material strength properties result in high
uncertainty of landslide stability analysis. Scenarios incorporating original soil cohesion showed stable conditions independent of saturation while cohesion-less scenarios indicated unstable scenarios independent or starting at low saturation levels. Mean soil cohesion scenarios revealed unstable conditions limited to high saturation levels. These saturation levels seem to be unlikely, however, unfavourable geologic conditions could result in high water pressures that develop between impermeable Feuerletten and clay layers within Rhätolias, reactivating deep-seated landslides.
Vegetation control is restricted to shallow landsliding. Roots of trees are limited to the upper 0.5 m due to unfavourable dry conditions at Rhätolias locations or unfavourable wet conditions, where Feuerletten are abundant. Root tensile strength is comparable between Norway spruce, Scots pine and European beech and root cohesion is mainly controlled by root area ratio. Therefore, shallow landsliding is highly unlikely at near-surface depth (0.3 m) where basal root cohesion provides sufficient stability. Below 0.5 m, lateral root cohesion can stabilize slopes even under high saturation without soil cohesion if gaps
between trees are avoided. Forest management can reduce landslide susceptibility by providing sufficient tree density and avoiding large scale harvesting.



## 7 Data availability

The DEM can be bought from Bayerisches Landesamt für Digitalisierung, Breitband und Vermessung (https://www.ldbv.bayern.de/produkte/3dprodukte/gelaende.html). The landslide inventory of Bavaria can be downloaded
from the Bayerisches Landesamt für Umwelt (https://www.lfu.bayern.de/umweltdaten/geodatendienste/pretty_downloaddienst.htm?dld=georisiken). The Third German National Forest Inventory (2011-12) is available from the Thünen-Institut (https://bwi.info/start.aspx). All data produced by the authors in this study is available at figshare (https://doi.org/10.6084/m9.figshare.20368464.v1).

**8 Author contribution**: DD designed the study and wrote the manuscript with contributions of TG and MP. All authors collected the ERT data. TG mapped the landslides and processed the ERT data as part of his BSc thesis. MP mapped tree location, measured tensile strength and root area ratios as part of her MSc thesis. DD re-analysed the landslide stability based on the theses by TG and MP.

**9 Competing interests**: The authors declare no competing interest.

## 10 Acknowledgements

The authors thank Fritz Maier and his team at Bayerische Staatsforsten Nordhalben for their support.

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
