# Peer review of "Geology and vegetation control landsliding on forest-managed slopes in scarplands"

_Earth Surface Dynamics, 2022_

## Author Comment (AC1)

**Reviewer Comments (**in black**), our response (**in blue**) and revised manuscript passages** (in dark orange**)**

**Reviewer 1:**

The authors present an analysis of two primary controls on slope stability in Northern Bavaria, Germany: geology and vegetation. The topic is important to protect life, property and infrastructure locally. The results also present possible contributions to our understanding of slope stability that would be applicable elsewhere.

I have two major comments that I believe will help to improve the paper.

Motivation/what's new? It is known that slope stability is influenced by both geologic and vegetation controls, the authors could better identify the knowledge gap and clearly illustrate how their study fills this knowledge gap. Specifically, the abstract jumps straight to the actions performed without motivating/asking a clear research question. The introduction only reaches a clear motivation towards the end – focusing on the vegetative controls of landsliding in shallow and low angled hillslopes. Is this the key knowledge gap (what controls shallow and deep landsliding on low angle slopes?) This should be made clearer in the abstract/intro to justify the study and used to better explain results in the discussion section.

Clearly define the two types of landslides and proposed controls. As written, the two types of landslides (deep and shallow) and the specific controls the authors investigate (geologic properties, vegetative root strength, respectively) are not clearly presented. Whereas some general background is appropriate in the introduction, there should be a sharpening of focus that clearly defines landslides of different depth, and the respective controls investigated in this study. It is initially unclear why the authors investigated tree root strength when the majority of landslides were all deeper than 2 m where no roots were found. The discussion of the tree root data similarly lacks focus and a take-home point because it is not clear why these data are included in the study.

Thank you for these valuable comments. We followed the reviewer's comments and differentiated deep-seated and shallow landslides from the beginning of the manuscript. We reworked the abstract, the introduction and the discussion to clarify the knowledge gaps and how we address these gaps. Following Reviewer 2, we also focused stronger on timescales that clarify how scarpland formation and associated geology affects even shallow landslides. On geological scale, scarpland geology preconditioned and prepared deep-seated landslides that are important processes shaping scarplands. As most slopes in our research area are affected by deep-seated landslides, these landslides can be reactivated and produce hazards or they precondition and prepare shallow landslides by setting the framework for these landslides (e.g. hillslope angle, sheared material). The geology influences the deep-seated landslides but also affected the rooting depth of trees by unweathered sandstone underlying permeable sand or saturated clayey soils above impermeable clays. Therefore, geologic conditions limited the effect of trees on shallow landsliding and enabled shallow landsliding on even low-inclined hillslopes. We reworked the abstract and stronger focused on the landslide types and their connection to geology and vegetation: "
[revised manuscript text omitted]

Additionally, there are useful tree data (DBH, age, stand density) that would add to the study.

We are not sure what the reviewer means. Is the reviewer referring to literature or our results. We did not measure DHB of all trees and only used DHB to select the trees where we sampled the roots. Furthermore, we assessed age qualitatively and discussed stand density for lateral root cohesion.

Specific comments:

Abstract        opening sentence is true, but what is knowledge gap paper attempts to fill? Clearly identify two types of landslides (shallow and deep) and the knowledge gaps on what controls these types of landslide on shallow slopes.

We sharpened the introduction and differentiate from the beginning deep-seated and shallow landsliding. We changed the abstract to: "Landslides are important agents of sediment transport, cause hazards and are key agents for the evolution of scarplands. Scarplands are characterized by high-strength layers overlying low-inclined landslide-susceptible layers that precondition and prepare landsliding on geological time scales. These landslides can be reactivated and their role in past hillslope evolution affected geomorphometry and material properties that set the framework for present-day shallow landslide activity. To manage present-day landslide hazards in scarplands, a combined assessment of deep-seated and shallow landsliding is required to quantify the interaction between geological conditions and vegetation that control landslide activity. For this purpose, we investigated three hillslopes affected by landsliding in the Franconian scarplands. We used geomorphic mapping to identify landforms indicating landslide activity, electrical resistivity to identify shear plane location and a mechanical stability model to assess the stability of deep-seated landslides. Furthermore, we mapped tree distribution, quantified root area ratio and root tensile strength to assess the influence of vegetation on shallow landsliding. Our results show that deep-seated landslides incorporate rotational and translational movement and suggest that sliding occurs along a geologic boundary between permeable Rhätolias sandstone and impermeable Feuerletten clays. Despite low hillslope angles, landslides could be reactivated when high pore pressures could develop along low-permeable layers. In contrast, shallow landsliding is controlled by vegetation. Our results show that rooted area is more important than species dependent root tensile strength and basal root cohesion is limited to the upper 0.5 m of the surface due to geologically controlled unfavourable soil conditions. Due to low slope inclination, root cohesion can stabilize landslide toes or slopes undercut by forest roads, independent of potential soil cohesion, when tree density is sufficient dense to provide lateral root cohesion. In summary, geology preconditions and prepares deep-seated landslides in scarplands, which set the framework of vegetation-controlled shallow landslide activity.

10            'rooted area' is supposed to be root area ratio?

Yes. We changed the sentence to: "Furthermore, we mapped tree distribution, quantified rooted area ratio and root tensile strength to assess the influence of vegetation on shallow landsliding. "

14         how do high pore pressures develop due to geologic conditions? Do you mean due to hydrologic conditions? Or increased pore pressure along low permeability boundary?

High pore pressure develops at the impermeable Feuerletten clays along low permeability boundaries. To clarify this, we changed the text to: "Our results show that deep-seated landslides incorporate rotational and translational movement and suggest that sliding occurs along a geologic boundary between permeable Rhätolias sandstone and impermeable Feuerletten clays. Despite low hillslope angles, landslides could be reactivated when high pore pressures could develop along low-permeable layers."

20         final 1-2 sentences of abstract would be stronger if they followed the 'two types of landslides' outlined above and distinguished how the mechanisms controlling slope stability are different in each (geology – forests)

We changed the abstract (as shown above) and differentiated stronger deep-seated and shallow landslides and their link to geology and vegetation.

Intro         why does the introduction start with a summary of sedimentary rocks? The paper is focused on geologic/vegetation controls on slope stability and as a reader I expect the principal topic to be one of those listed in the title.

We adapted the introduction and start now with scarplands and how they influence deep-seated landslides on geological time scales. This introduction is necessary as geology preconditions landslide movement. Afterwards, we explain why the deep-seated landslides can be reactivated and why deep-seated landslides set the framework for shallow landslides effected by vegetation. Shallow landsliding is affected by the geology as soil conditions influence the rooting depth of trees and, therefore, influences root cohesion.

43         Also Schmidt, Roering, Ziemer, Terwilliger & Waldron.

Added.

55         also Ziemer (https://www.fs.usda.gov/treesearch/pubs/8693)

Added.

61-62 Ziemer and Terwilliger and Waldron ([https://pubs.geoscienceworld.org/gsa/gsabulletin/article-abstract/103/6/775/182576/Effects-of-root-reinforcement-on-soil-slip](https://pubs.geoscienceworld.org/gsa/gsabulletin/article-abstract/103/6/775/182576/Effects-of-root-reinforcement-on-soil-slip))

Added.

40-62         tighten language as there is some repetition

We rewrote this section completely: "As deep-seated landslides were important in shaping scarplands, they changed the geomorphometry of hillslopes (e.g. inclination) and sheared material and, therefore,

precondition and prepare present-day shallow landslides. Shallow landslides are characterized by soil material <2 m deep moving downslope in a flowing, sliding or complex type of movement (Sidle and Bogaard, 2016; Vergani et al., 2017). Forests can affect shallow landsliding mechanically and hydrologically (Vergani et al., 2017). They can reduce soil moisture by interception and evaporation, suction and transpiration as well as infiltration and subsurface flow (Sidle and Bogaard, 2016; Vergani et al., 2017). Mechanically, forests can reinforce soil by roots (Wu, 1984; Phillips et al., 2021), roots and stems can induce buttressing (Vergani et al., 2017) and anchoring and trees can increase normal force on slopes (Ziemer, 1981; Terwilliger and Waldron, 1991; Selby, 1993; Schmidt et al., 2001; Roering et al., 2003). In forest management, the protective function of forests has been considered for a long time in high mountain regions (Dorren et al., 2005; Bischetti et al., 2009). However, forestry is not only affected by landslide activity, which causes damage to roads and loss of timber (Sidle and Ochiai, 2006), but also has a considerable impact on slope stability through changing the characteristics of forests in sliding-prone areas (Phillips et al., 2021). Root reinforcement of slope stability declines after logging operations (Ziemer, 1981; Schmidt et al., 2001; Vergani et al., 2017) and forestry roads enhance landsliding through undercutting slopes (Borga et al., 2005; van Beek et al., 2008). Changes in tree species composition and tree density have also an impact on the root reinforcement in forests (Roering et al., 2003; Genet et al., 2008)."

62-67          Good motivation for study – but should also clearly distinguish between shallow and deep and the controls of geology and vegetation. This reasoning should be in abstract

We followed the comment of the reviewer and added the motivation to the abstract: "Scarplands are characterized by high-strength layers overlying low-inclined landslide-susceptible layers that precondition and prepare landsliding on geological time scales. These landslides can be reactivated and their role in past hillslope evolution affected geomorphometry and material properties that set the framework for present-day shallow landslide activity. To manage present-day landslide hazards in scarplands, a combined assessment of deep-seated and shallow landsliding is required to quantify the interaction between geological conditions and vegetation that control landslide activity."

117          cite RMS from previous investigations and briefly summarize what was found

We used the study by Lapenna et al. (2005), which was cited in the review paper by Perrone et al. (2014) and adjusted the sentence: "Model results showed a low root mean square (RMS) error between 5.3 and 5.4% for Putzenstein and Weinreichsgrab and an increased RMS error of 12.1% at Fürstenanger. RMS values are comparable to previous investigations identifying shear planes at clayey sand layers in the Flemish Ardennes (Van Den Eeckhaut et al., 2007; RMS 4.1 - 14.5 %) or clay layers in the Apennine (Lapenna et al., 2005; RMS 2.3 - 15.1 %)."

127          dead/cut trees were excluded, but dead/cut trees continue to provide strength until they rot away. See Ziemer:

We changed the text to: "Dead and cut trees were excluded as the influence of roots on cohesion decreases with ongoing decomposition (Vergani et al., 2014; Zhu et al., 2020) until trees rot away (Ziemer, 1981; Ammann et al., 2009)." We added also information to the discussion section: "Our analysis excluded dead or harvested trees that can provide additional root cohesion until they rot away (e.g. Ammann et al., 2009; Vergani et al., 2017), therefore, we eventually underestimate both basal and lateral root cohesion."

140          only 1 species (Scots Pine) was measured in this study and roots .

We clarified this point and changed the text to: "To measure root tensile strength of Scots pine, root samples with different diameters and a minimum length of 10 cm were extracted. […] A power-law between root tensile strength and root diameter $d$ can be established for Scots pine:

$$T_r(d) = \alpha d^{-\beta} \tag{5}$$

with α and β are empirical constants depending on species. In addition, power laws for Norway Spruce (18.10 $d^{-0.72}$, r² = 0.52) and European Beech (41.57 $d^{-0.98}$, r² = 0.65) established by Bischetti et al. (2009) were used in our analysis."

195          should gs represent the saturated bulk density of the soil?

We found a measured value of specific weight for Feuerletten and Rhätolias and changed the sentence to: "..with $\gamma_s$ is the specific weight of Feuerletten or Rhätolias in the order of 21 kN m⁻³ (Boley Geotechnik, 2018) and $B_i$ is the width of each slice." We adapted all landslide stability models, however, modelled factor of safety only changes slightly and the observed pattern not at all.

215          goal to 'test if root cohesion would be sufficient to stabilize the soil' of shallow landslides should be mentioned in the introduction.

We added this information to the introduction: "Furthermore, we (2) test if vegetation-induced root cohesion can stabilize shallow landslides occurring on deep-seated landslides. For this reason, we mapped tree distribution, quantified root cohesion and applied a slope stability model."

Fig 2 legend  'transekt' should be 'transect', since Rhätolias-Feuerletten boundary is so important, consider changing color to make it stand out.

Thanks for the comment. We changed the spelling error and highlighted the Rhätolias-Feuerletten boundary in red with an increased line width.

Figure 4      explain in legend the criteria used to identify failure plane boundary – I had to go back and search to find line 122 about Figure S3 and the identified shear plane depth

Thanks for the comment. We find it difficult to add this information to the figure and we added the information to the figure caption: "Geoelectric models and landslide forms at (a) Putzenstein, (b) Weinreichsgrab and (c) Fürstenanger. Failure plane depth was derived from vertical resistivity decrease in order of one to two magnitudes. For detailed derivation see Figure S3 in the Supplementary Information. F highlights location of forest roads."

Figure 5     legend should include scarps, caption should tell reader locations of panels a, b, c, referring to the maps in figure 2. Fürstenanger is the only location with a spatial pattern in species – with Scots Pine concentrated near headscarp. Is this important?

We added more information to the legend (see below) and added more information to the figure caption on the location of the ERT transects in Figure 2, which is Figure 3 in the revised version. Figure caption changed to: "Figure 6: Mapped trees with height above 4 m in up to 5 m distance to the ERT transects (Fig. 5) at (a) Putzenstein, (b) Weinreichsgrab and (c) Fürstenanger. The locations of ERT transects are shown in Fig. 3."

[Figure]

The reviewer is right, the headscarp of the Fürstenanger landslide shows a concentration of Scots Pine. The steep parts of slopes are protected forest to reduce erosion. As the roots only reach half a meter deep, the trees will have no effect on the movement of the deep-seated landslide. In addition, the headscarp is not well accessible with forest machines and forestry activities are reduced in this area. In summary, the concentration is more a result of forest management that has not changed the tree composition yet but has no influence on landslide activity.

293        This sentence is not clear. What does 0.19 refer to?

The value refers to RAR. We changed the text to: "For Norway spruce, mean root area ratio decreased from the surface to 0.5 m with values between 0.19 and 0.2 % at 0 to 0.2 m depth, 0.04 % at 0.2 to 0.4 m depth and 0.005 % between 0.4 and 0.5 m depth (Fig. 8a)."

Figure 6        why do the authors plot root diameter against tensile strength in MPa instead of against tensile force at failure? I recommend including the previously published data to show the stated similarities with other species

In our literature review, we found authors doing both, plotting root diameter against tensile force at failure or root diameter against tensile strength. We follow the approach by Bischetti et al. (2009) and plotted the data in the same way. Also Genet et al. (2005), Ji et al. (2012) and others preferred root diameter versus tensile strength. As we used the power law between root diameter and tensile strength to derive root cohesion, we thought it is logical to present the relationship in that way in our figure. The reviewer is right that plotting similarities to other species especially Norway spruce and European beech improves the figure and enables a visual comparison between species. We used the power laws by Bischetti et al. (2009), however, data of this study is unfortunately not open accessible. Instead of plotting the data, we plotted the derived power law as previously done by several papers.

[Figure]

Figure 7: Tensile strength plotted versus root diameter for Scots pine compared to power laws derived for European Beech and Norway Spruce.

317       unclear sentence, instead of stating 'get' unstable, I suggest 'become' unstable or fall below FoS of 1.

We changed the sentence to: "The Weinreichsgrab landslide became instable when saturation increases above 0.8 in the upper slice height scenario (Fig. 9b)."

Figure 7      why are there no data for 0.5 cm depth in the European Beech? And, are the authors sure there are no roots deeper than 0.5 m that would add tensile strength to the soil?

For European beech, our data showed no roots between 0.4 and 0.5 m. We understand that not displaying the data points as 0 could also implicate that no data was available to other reasons (e.g. technical reasons). We changed this and plotted 0 % RAR in Fig. 7c (Figure 8c in the revised version) and 0 kPa root cohesion in Fig. 7f (Figure 8f in the revised version). Field measurements showed that there were no roots deeper than 0.5 m for all three species, which we have not expected before. We suggest that the surprisingly low rooting depth is a result of the geologic conditions. Rhätolias sandstone is partly not weathered and when weathered the water holding capacity is very low resulting in high resistivities in the ERT. Due to this unfavourable combination, the rooting depth is reduced in the upper slope parts. In the lower parts affected by Feuerletten, the impermeable clay layers result in very wet conditions expressed as low resistivities in the ERT. Consequently, roots add no tensile strength to the soil below 0.5 m. We added a new Figure 1 as suggested by Reviewer 2 including a schematic representation of the geology and soil pits showing soil conditions and roots in Rhätolias sandstone and Feuerletten clay.

[Figure]

**Figure 8: Root area ratio plotted against depth for (a) Norway spruce, (b) Scots pine and (c) European beech. Root cohesion plotted against depth for (d) Norway spruce, (e) Scots pine and (f) European beech. Red dots highlight mean RAR or root cohesion.**

[Figure]

**Figure 1: (a) Geological profile of investigated slopes in the Franconian Alb. Soil pits showing the upper 0.5 m of soil developed in (b) Rhätolias sandstone and (c) Feuerletten clay.**

Fig 8 caption 'We assume an angle of internal friction of 8.4°. We vary cohesion between…

Text changed to: "Factor of safety models for the reactivation of the landslides at (a) Putzenstein, (b) Weinreichsgrab, and (c) Fürstenanger. We assume an angle of internal friction of 8.4°. We vary cohesion between 28.6 kPa (blue scenario), 8.5 kPa (yellow) and 0 kPa (green)."

334        'All these locations are underlain by...'

Changed.

Figure 9        I like the figure, the different colors are hard to see.

Thanks. We increased the line width to improve visibility.

[Figure]

Figure 10: Factor of safety for full-saturated conditions with a residual angle of friction of 8.4° plotted against cohesion scenarios ranging from no cohesion to 10 kPa for (a) translational landslides at road cuts and (b) landslide toes. Line style highlight the depth of shear plane ranging between 0.3 m and 1.5 m. Line colour in (a) refer to Putzenstein (black) with a slope angle of 13°, Weinreichsgrab and Fürstenanger (both blue) with slope angles of 12°. Line colour in (b) refer to Putzenstein (black) with a slope angle of 11°, Weinreichsgrab (blue) with a slope angle of 9° and Fürstenanger (green) with a slope angle of 6°.

349        sentence structure 'Of the 125 observed landslides, 95% occurred at the R-F boundary...'

Changed.

360        'In between the lower high-resistivity cells...'

We changed high resistant or high-resistant to high resistivity or high-resistivity in this manuscript.

362        'The lower part of the landslide was characterized by flat topography, low-resistivity areas...'

Changed.

392        unclear what this sentence is trying to communicate 'Water can move laterally...'

Rhätolias and Feuerletten are both inhomogeneous and contain clay layers. Due to tectonic activity, fractures were observed within Rhätolias that can enable infiltration through clay layers according to Wilfing et al. (2018). When the water moves laterally slope downward, the water can be trapped between impermeable clay of the  Feuerletten and a clayey layer in the overlying Rhätolias. This situation can increase the pore pressure as observed by several studies. We change the sentence to clarify this mechanism: "However, Rhätolias has impermeable clay layers (Boley Geotechnik, 2018) and

tectonic-induced fractures can increase water infiltration through these clay layers (Wilfing et al., 2018). Therefore, water can be trapped between clay layers in Rhätolias and clay layers in underlying Feuerletten, which can cause hydrostatic pressures equal to high saturation levels (Rogers and Selby, 1980; Selby, 1993)."

444          what effect might lateral root cohesion have on such a broad landslide?

Lateral root cohesion can prevent the initiation of shallow landslides or limit the size. We changed the sentence to: "When shear planes exceed rooting depth, lateral root cohesion can have a stabilizing effect (Schwarz et al., 2010b) by affecting the onset and size of shallow landsliding (Schmidt et al., 2001; Roering et al., 2003) as indicated by tensed roots observed at Putzenstein (Fig. 4b)."

---

## Author Comment (AC2)

**Reviewer Comments (**in black**), our response (**in blue**) and revised manuscript passages** (in dark orange**)**

**Reviewer 2:**

In this paper the authors analyzed the geologic and vegetation control on deep and shallow landslides in the scarplands of Southern Germany. Their objective was to understand geologic conditions and the role forest management might play in helping slope stability, especially when these slopes face changing vegetation and hydrologic conditions in a changing climate.

The interplay between vegetation and sub-surface conditions to better understand regional landslide hazards is a long-studied and important topic of interest. This paper presents many useful datasets from Northern Bavaria (e.g. ERT data for several large landslides, root strength from Scots Pine, detailed landslide mapping, and soil strength properties). However, the paper would greatly improve with a more clearly defined hypotheses and motivation, which are then justified by the data and the discussion. As written, this is almost two papers: 1) about rooting strength and controls on recent shallow landsliding, and 2) about modelling the unique scarpland geology that leds to pervasive large, deep landslides that are now seemingly stable.

The reviewer is right, we failed to explain sufficiently our motivation and why we investigate both deep-seated and shallow landslides. We adapted the abstract, introduction and discussion. We present from the start of the manuscript on what the differences between deep-seated and shallow landslides are and on what time-scales these processes occur. On geological time scale, scarpland formation with alternating sedimentary layers precondition deep-seated landslides as high-strength layers overlay weak sedimentary layers. Fluvial erosion prepared deep-seated landsliding by exposing the weak layers and the landslides were caused by different potential triggers. The landslide processes changed topography (e.g. slope angle) and resulted in sheared material. On present-day scale, the deep-seated landslides set the framework for current landslide processes. These include a re-activation of deep-seated landslides or shallow landsliding on top of the deep-seated landslides. The shallow landsliding is controlled by vegetation especially root cohesion. The root cohesion is influenced by the soil thickness affecting root area ratio, which is controlled by geologic conditions. The upper hillslopes consist of Rhätolias which weathered into sandy permeable soil underlying by unweathered sandstone that restrict root depth to the upper 0.5 m. In the lower part of hillslopes, Feuerletten are abundant and soils are low permeable clay that induced wet conditions and, therefore, limited rooting depth. The sedimentary layering of the scarplands controls how vegetation affects hillslope stability and shallow landsliding. In summary, geology controlled past deep-seated landsliding that was involved in the formation of scarplands and controls present-day deep-seated and shallow landsliding. To highlight this interaction, we assessed both landslide types in one paper rather than two papers addressing each type individually.

A few major comments:

Abstract – This does not flow well and lacks a clear framing of the authors' hypotheses. The abstract mentions many methods but it is not clear how all these methods intersect.

The reviewer is correct. To improve the motivation and clarify the use of methods, we rewrote the abstract. New abstract: "Landslides are important agents of sediment transport, cause hazards and are key agents for the evolution of scarplands. Scarplands are characterized by high-strength layers overlying low-inclined landslide-susceptible layers that precondition and prepare landsliding on geological time scales. These landslides can be reactivated and their role in past hillslope evolution

affected geomorphometry and material properties that set the framework for present-day shallow landslide activity. To manage present-day landslide hazards in scarplands, a combined assessment of deep-seated and shallow landsliding is required to quantify the interaction between geological conditions and vegetation that control landslide activity. For this purpose, we investigated three hillslopes affected by landsliding in the Franconian scarplands. We used geomorphic mapping to identify landforms indicating landslide activity, electrical resistivity to identify shear plane location and a mechanical stability model to assess the stability of deep-seated landslides. Furthermore, we mapped tree distribution, quantified root area ratio and root tensile strength to assess the influence of vegetation on shallow landsliding. Our results show that deep-seated landslides incorporate rotational and translational movement and suggest that sliding occurs along a geologic boundary between permeable Rhätolias sandstone and impermeable Feuerletten clays. Despite low hillslope angles, landslides could be reactivated when high pore pressures could develop along low-permeable layers. In contrast, shallow landsliding is controlled by vegetation. Our results show that rooted area is more important than species dependent root tensile strength and basal root cohesion is limited to the upper 0.5 m of the surface due to geologically controlled unfavourable soil conditions. Due to low slope inclination, root cohesion can stabilize landslide toes or slopes undercut by forest roads, independent of potential soil cohesion, when tree density is sufficient dense to provide lateral root cohesion. In summary, geology preconditions and prepares deep-seated landslides in scarplands, which set the framework of vegetation-controlled shallow landslide activity."

Introduction – The end of the introduction seems to describe the overall motivation (looking at role of vegetation on low slopes with landslides) that should be made clear much earlier and returned to throughout. For this section, I suggest shortening the forestry summary and expanding on the scarpland morphology and differences between shallow/deep landslides to introduce your hypotheses. With a clear motivation and hypotheses, the methods can then be justified.

Additionally, the paper would be strengthened by describing the different timescales of interest (e.g. recent land-use/forestry management and shallow landslides, longer-term climate shifts possibly leading to reactivation or initiation of deep landslides, timing of original (now "fossil") landslides).

We followed the recommendations of the reviewer and rewrote large parts of the introduction. We introduced the differences between deep-seated and shallow landslides and different timescales from the beginning on, shortened the forest summary and expanded scarpland morphology: "Landslides are important agents of sediment transport, cause hazards and are key agents for the evolution of scarplands. On geological scale, sedimentary deposition in terrestrial or marine environments resulted in alternating layers of different rock strength with varying inclination (Duszyński et al., 2019), which preconditions slope stability (McColl, 2022). Horizontal layering promotes the formation of plateaus, while tilted layers create cuestas (Young et al., 2000; Duszyński et al., 2019). Due to the differences in rock strength and resulting different efficacy of erosive processes, scarplands are characterized by high-strength layers overlying weaker sedimentary layers (Duszyński et al., 2019). Tectonic processes can increase slope height or slope steepness and erosion (e.g. by rivers) can undercut hillslopes and expose weaker sedimentary layers, which act as potential failure surfaces, and, thereby prepare landslide processes (McColl, 2022). Landslides can be caused by a wide range of triggers including e.g. rapid increase in pore water pressure by rainfall and/or snowmelt, loading of slope by precipitation or vegetation (McColl, 2022). The tilting of sedimentary layers controls the landslide type in scarplands. On frontscarps, sediment layers dip into the slope (Duszyński et al., 2019) and landslides in form of rockfall (e.g. Glade et al., 2017) or deep-seated landslides (e.g. Jäger et al., 2013) are abundant. In contrast, sedimentary layers dipping out of the slope characterize backscarps (Schmidt and Beyer,

[revised manuscript text omitted]

Discussion – As written, this section seems to raise more questions than it answers. For example, why such a focus on tree roots when the landslides analyzed are much much deeper than the rooting depth? Or, for stability modelling, why not model the geologic and hydrologic conditions needed for the original failures to test the importance of scarpland geology (e.g. Perkins et al., 2017)? This section could be strengthened by clearly describing: what is novel from this study, how low angle hillslopes compare to steep vegetated hillslopes, and intersection of geology and deep landslides with vegetation and shallow landslides.

Thank you for these valuable comments. The model approach by Perkins et al. (2017) combining the landslide stability model Scoops3D with the hydrological model VS2Dt sounds really interesting and would be suited if we want to model more accurately landslide stability of deep-seated landslides maybe in a future manuscript. From our point of view, a new model approach would not provide any more valuable information for this paper, as the purpose of this paper is a different one. We want to incorporate as well the geological control on current shallow landslides affected by vegetation. As the reviewer commented correctly in previous comments, we failed to provide a clearer motivation and, therefore, we revised the abstract and introduction to clarify our motivation and used set up. We revised the discussion section to clarify what is novel of our paper. In the first section of the discussion, we focus on geologic control on deep-seated landsliding. We revised the section on the role of deep-seated landslides for scarpland formation:

"The combination of high-permeable Rhätolias above Feuerletten controls deep-seated landsliding. Of the 125 observed landslides in our research area, 95 % occurred at the Rhätolias-Feuerletten boundary (Fig. 2b), which suggests that Feuerletten play a key role in landsliding. The Feuerletten possess a lower angle of internal friction than Rhätolias (Table 1) and cohesion of these clays is susceptible to saturation. Previous landslide inventories of the Franconian Alb support this role of Feuerletten in the North-Bavarian scarplands, where Feuerletten were responsible for an inappropriate high proportion of landslides (Kany and Hammer, 1985). Kany and Hammer (1985) assumed that most landslides were fossil and occurred under past climatic conditions, however, they suggested that these deep-seated landslides could be reactivated due to anthropogenic impacts as road cutting and forestry. The observed movement of the Thurnau landslide affecting the highway (Fig. 2b; Wilfing et al., 2018) supports the argument of potential reactivation."

We clarified the purpose of the application of the ERT and what was the major outcome: "The ERT enabled the identification of the shear plane location and suggested that landslides are complex with rotational and translational movement. The Putzenstein landslide revealed a hummocky topography (Fig. 3a) and the ERT showed three high-resistivity cells with resistivities up to 4,000 Ohm m and a thickness between 7 and 18 m located above low-resistivity bodies at transect length between 5 and 70 m, 70 to 195 m and 232 to 315 m (Fig. 5a). Pürckhauer drillings revealed fine and silty sand in the upper 1 m. We interpret these cells as dry Rhätolias above wet Feuerletten. The form of these cells and the hummocky topography indicate three rotational slabs. In between the lower high-resistivity cells, low resistivities indicate a water-saturated rotational slab (Fig. 5a). The lower part of the landslide was characterized by a flat topography, low-resistivity areas, and near-surface clay material. Therefore, we interpret this landslide part as a translational slide within the Feuerletten. The Weinreichsgrab landslide revealed a similar pattern of three high-resistivity cells within hummocky terrain with near-surface silty sand followed by flat terrain with low resistivities and near-surface clay (Fig. 5b). These results indicate three rotational slabs and one translational slab at the toe of the landslide. In contrast,

[revised manuscript text omitted]

We moved the forestry management to an additional chapter called "Potential impacts of forestry activity on future shallow landsliding".

Specific comments:

ABSTRACT

13: does low slope refer to the geologic contact or hillslope angle (likely hillslope, but confusing after talking about dipping angles)? Does high pore pressure refer to a measured, modelled, or inferred point?

AND

14: geologic conditions should be hydrologic

AND

18: why is European beech specifically helpful to landslide stability? If making this recommendation, include results leading to this conclusion.

We rewrote the abstract. We reshaped the sentence on used techniques and from the revised sentence it should be clear that we did not measure any pore pressures: "For this purpose, we investigated three hillslopes affected by landsliding in the Franconian scarplands. We used geomorphic mapping to identify landforms indicating landslide activity, electrical resistivity to identify shear plane location and a mechanical stability model to assess the stability of deep-seated landslides. Furthermore, we mapped tree distribution, quantified root area ratio and root tensile strength to assess the influence of vegetation on shallow landsliding." Furthermore, we changed "slope" to "hillslope angle" to clarify that we are not referring to dipping angle of the sedimentary layers. In addition, we followed the comment of Reviewer 1 and substituted "geological conditions" with "along low-permeable layers". New sentence is: Despite low hillslope angles, landslides could be reactivated when high pore pressures could develop along low-permeable layers."

INTRODUCTION:

24: "sedimentary origin" should be specified as "scarplands"

AND

24-25: jump into very detailed geology and landslide classifications without setting up overall objectives

We rewrote the entire section of the introduction section and explained in more detail how scarpland formation or properties precondition and prepare deep-seated landslides. We also set up the objective of studying deep-seated landslide much earlier. "Landslides are important agents of sediment transport, cause hazards and are key agents for the evolution of scarplands. On geological scale, sedimentary deposition in terrestrial or marine environments resulted in alternating layers of different rock strength with varying inclination (Duszyński et al., 2019), which preconditions slope stability (McColl, 2022). Horizontal layering promotes the formation of plateaus, while tilted layers create cuestas (Young et al., 2000; Duszyński et al., 2019). Due to the differences in rock strength and resulting different efficacy of erosive processes, scarplands are characterized by high-strength layers overlying weaker sedimentary layers (Duszyński et al., 2019). Tectonic processes can increase slope height or slope steepness and erosion (e.g. by rivers) can undercut hillslopes and expose weaker sedimentary layers, which act as potential failure surfaces, and, thereby prepare landslide processes (McColl, 2022). Landslides can be caused by a wide range of triggers including e.g. rapid increase in pore water pressure by rainfall and/or snowmelt,  loading of slope by precipitation or vegetation (McColl, 2022). The tilting of sedimentary layers controls the landslide type in scarplands. On frontscarps, sediment layers dip into the slope (Duszyński et al., 2019) and landslides in form of rockfall (e.g. Glade et al., 2017) or deep-seated landslides (e.g. Jäger et al., 2013) are abundant. In contrast, sedimentary layers dipping out of the slope characterize backscarps (Schmidt and Beyer, 2003; Duszyński et al., 2019), where landsliding processes comprise cambering (Hutchinson, 1991), block gliding (Young, 1983), lateral spreading (Spreafico et al., 2017) or deep-seated sliding processes (Pain, 1986; Schmidt and Beyer, 2003). Geologic conditions precondition landsliding and the formation of scarplands on geological scale. On present-day, reactivation of  deep-seated landslides by geomorphic and anthropogenic processes (McColl, 2022) cause hazards to communities living in scarplands (Thiebes et al., 2014; Wilfing et al., 2018), therefore, an understanding of geologic controls on landsliding is required to analyse slope stability for hazard management."

36: Using only a depth cutoff is a little misleading, typically shallow=landslide rooted in soil and deep=landslide rooted in rock.

We followed the reviewer's comment and added the word soil to the definition of shallow landslides: "Shallow landslides are characterized by soil material <2 m deep moving downslope in a flowing, sliding or complex type of movement (Sidle and Bogaard, 2016; Vergani et al., 2017)." We highlighted that scarpland formation resulted in different rock layers that affect deep-seated landslides.

50: change "therefore" to "and"

This criticized sentence was deleted in the revision process.

63: If this is the motivational framework, introduce early on and include in abstract. Frame your hypotheses or research questions based on this motivation. For example, in low slope scarplands do you expect vegetation to have more or less influence than steep mountains?

We revised the motivation and framed our research questions better on the motivation: "As geological conditions control deep-seated landslide activity on geological scale that set the framework for shallow landslides in scarplands on present-day scale, there is a need to understand how landslide historicity affects current deep-seated and shallow landslide activity. As climate change affects forests (e.g. Seidl et al., 2017) and alters landslide activity (e.g. Crozier, 2010), combined forestry management and hazards approaches on shallow landslides (Phillips et al., 2021) should be extended by incorporating geological controls in scarplands. We revised our objectives to link these closer to the motivation: "In this study, we aim to (1) quantify the relation between deep-seated landslides and geology in the Franconian Alb and estimate if landslides can be reactivated by hydrologic conditions. For this purpose, we extended a landslide inventory and compared landslide occurrence to geology. On three landslides, we applied electrical resistivity tomography (ERT) to identify shear plane depth and modelled hillslope stability with different water level scenarios. Furthermore, we (2) test if vegetation-induced root cohesion can stabilize shallow landslides occurring on deep-seated landslides. For this reason, we mapped tree distribution, quantified root cohesion and applied a slope stability model. Our results aim to improve forest management practices to reduce landslide occurrence in the Franconian Alb."

METHODS:

125-126: Studies show that significant root strength can persist for up to ~10 years (e.g. Ammann et al., 2009-Norway Spruce). What is age of trees vs. age of landslides?

The deep-seated landslides were formed probably under past climatic conditions. The shallow landslides are recent landslides and it is hard to establish an age. The trees are definitely older as they are bent or tilted or show tensed roots, which are all effects of landslide activity. We added the information of the Ammann paper to the method section: "Dead and cut trees were excluded as the influence of roots on cohesion decreases with ongoing decomposition (Vergani et al., 2014; Zhu et al., 2020) until trees rot away (Ziemer, 1981; Ammann et al., 2009)." We also added information to the discussion: "Our analysis excluded dead or harvested trees that can provide additional root cohesion until they rot away (e.g. Ammann et al., 2009; Vergani et al., 2017), therefore, we eventually underestimate both basal and lateral root cohesion."

127: insert "…selected 15 individual free-standing…"

Done.

128: remove "at 15 trees"

Done.

182: how was this material collected? Does this include bedrock material? Or just landslide material?

The material was collected by the company Boley using 35 boreholes with between 30 and 240 m depth, in total 1700 m were drilled on landslide material and neighbouring bedrock not affected (yet)

by the landslide. The aim of the investigation was to find an alternative route for the affected highway. We changed the text to: "Mechanical strength parameters of Feuerletten and Rhätolias were quantified using approximately 90 circular, direct and triaxial tests on materials derived from 35 boreholes on the Thurnau landslide affecting the highway (Fig. 2B) and surrounding bedrock (Boley Geotechnik, 2018; Wilfing et al., 2018)."

RESULTS:

119: what do you mean by "follow no expositional pattern"?

We meant that the landslides show no preference of exposition. If landslides are climatic driven such as driven by permafrost, which would be in our case more than 20,000 years ago, the landslides on south-facing slopes show usually a different pattern then on north-facing slopes. As this is not the objective of the manuscript, we deleted the part on expositional pattern.

DISCUSSION:

352-353: why not try to model this instead? Similar to Perkins et al., 2017

Thank you for this comment and the paper, which applied very interesting models. Our study focuses on field measurements with a modelling component. Applying the Scoops3D model in combination with VS2Dt model would be a very useful for future work maybe in form of PhD project. However, as the reviewer mentioned, this paper is very dense and maybe two papers and incorporating two new model approaches would completely shift the focus more to the deep-seated landslides, which is not the aim of our study. We will reframe the motivation to make this clearer.

376-377: Why such a focus on trees then? Why not focus on the mechanics of these deep landslides using stability modeling and ERT results?

The initial motivation of the study was to investigate the role of trees on landsliding and provide a recommendation on tree selection for hazard management. During our study, it became clearer that geology affects not only the deep-seated landslides but also the shallow landslides as geologic conditions influence soil and limited rooting depth in our case. We think that this is the interesting point for our study but also for scarplands. You cannot assess vegetation influence without incorporating geological conditions as geology preconditions landslides. We made this clearer in the abstract, introduction and discussion of the manuscript.

378: Where on the landslide? Just deposit/mobile material?

Both. See our detailed answer above (Response to line 182).

390-395: There are a lot of hypotheses presented here without much to back-up conclusions. Seems like the more interesting modeling problem if the authors want to understand the role of scarpland geology

Here, we disagree. We could apply a more sophisticated model as the reviewer suggested. However, the material properties vary which results in a large variation of potential stability scenarios. We could improve the hydrological part of the modelling but it will not solve the problem of constraining the material variation. We also would only investigate the role of deep-seated landslides on scarpland formation, however, or focus, which we not clearly enough explained, was how the geological framework preconditioned and resulted in deep-seated landslides that on present-day affect forestry.

426-427: what are the depths of shallow landslides occurring on these larger landslides?

We added the information to the discussion section: "Tensed roots at Putzenstein (Fig. 4a-c) and bent or tilted trees at Weinreichsgrab (Fig. 4f) indicate soil creep or shallow landsliding in the upper 1 to 1.5 m of Feuerletten clay (Fig. 3a-b)."

FIGURES:

NEW FIG(s): Suggest adding schematic of geology and/or typical slope profile, and photo of typical soil pit with roots.

Thank you for this comment. We added a geological sketch and photos of soil pits. Slope transects can be derived from the ERT transects, where topography was incorporated.

[Figure]

**Figure 1: (a) Geological profile of investigated slopes in the Franconian Alb. Soil pits showing the upper 0.5 m of soil developed in (b) Rhätolias sandstone and (c) Feuerletten clay.**

FIG 5: missing symbology

We added the symbology (see below).

| Landslide landforms | Mapped species | RAR tree species | Geophysical transect |
|---|---|---|---|
| Main scarp | ○ Birch | ★ European beech | Forest road |
| Secondary scarp | ● European beech | ★ Scots pine | Countour line 10 m |
| Fissure | ● European larch | ★ Norway spruce | Tree mapping |
| Slope depression | ● Norway spruce | | Rhätolias-Feuerletten boundary |
| Front | ● Scots pine | | |
| Main scarp, recent | ● Willow | | |
| Secondary scarp, recent | | | |
| Fissure, recent | | | |
| Landslide deposit | | | |

---

## Editor Decision (ED1)

Abstract and introduction:
Is it possible to somehow clarify the difference between what is meant by precondition vs. prepare? It seems that precondition refers to the geologic structure that leads to water and associated pore pressure accumulation. That then leaves a reader wondering what is meant be prepare and whether that is redundant?

Please change low-permeable to low permeability. Please ensure that references to permeability or hydraulic conductivity are consistent throughout the manuscript.

Fig1a: Please clarify in the caption why the geologic profile has the 2D shape that it does. Is it meant to be 2D or simply distorted to represent a slope?

L489: In dry locations?

L509: Correct to "at depths too deep for tree roots"

L512: Is it possible to be more specific about what is meant by "low" or "high" saturation levels and distinguish between thickness of water saturation (i.e. height of water table) vs water saturation at a given depth?

---

## Author Response (AR2)

**Editor Comments (**in black**), our response (**in blue**) and revised manuscript passages** (in dark orange**)**

Abstract and introduction:
Is it possible to somehow clarify the difference between what is meant by precondition vs. prepare? It seems that precondition refers to the geologic structure that leads to water and associated pore pressure accumulation. That then leaves a reader wondering what is meant be prepare and whether that is redundant?
We understand this issue. Preconditioning factors influence hillslope stability but are unchanging over time as geology. In contrast, preparing factors change the stability over time for example a river undercutting a slope or deforestation and change hillslope stability from stable to unstable but without triggering the failure. Due to the word limit, we could not adapt the abstract without large changes. Therefore, we added a sentence to the introduction to clarify the difference between preconditioning and preparing factors. "Preconditioning factors influence hillslope stability and are temporarily unchanging, while preparing factors reduce hillslope stability over time to an actively unstable state."

Please change low-permeable to low permeability. Please ensure that references to permeability or hydraulic conductivity are consistent throughout the manuscript.
Done.

Fig1a: Please clarify in the caption why the geologic profile has the 2D shape that it does. Is it meant to be 2D or simply distorted to represent a slope?
The profile is distorted to represent a slope. We changed the figure caption to: " (a) 2D slope profile with the major geological units in the Franconian Alb."

L489: In dry locations?
Changed.

L509: Correct to "at depths too deep for tree roots"
Done.

L512: Is it possible to be more specific about what is meant by "low" or "high" saturation levels and distinguish between thickness of water saturation (i.e. height of water table) vs water saturation at a given depth?

We changed the text to clarify the differences between saturation as a result of increased height of the water table and high saturation or better high water pressure as a result of water enclosed between impermeable layers. Text changed to: "Scenarios incorporating original soil cohesion showed stable conditions independent of saturation while cohesion-less scenarios indicated unstable scenarios independent or starting at low height of water table. Mean soil cohesion scenarios revealed unstable conditions limited to high saturation levels during increased heights of water table. These saturation levels seem to be unlikely, however, unfavourable geologic conditions could result in high water pressures that develop between impermeable Feuerletten and clay layers within Rhätolias, reactivating deep-seated landslides."